# Genetic Effect and Growth Curve Parameter Estimation under Heat Stress in Slow-Growing Thai Native Chickens

**DOI:** 10.3390/vetsci8120297

**Published:** 2021-11-29

**Authors:** Wuttigrai Boonkum, Monchai Duangjinda, Srinuan Kananit, Vibuntita Chankitisakul, Wootichai Kenchaiwong

**Affiliations:** 1Department of Animal Science, Faculty of Agriculture, Khon Kaen University, Khon Kaen 40002, Thailand; wuttbo@kku.ac.th (W.B.); monchai@kku.ac.th (M.D.); srinka@kku.ac.th (S.K.); vibuch@kku.ac.th (V.C.); 2Network Center for Animal Breeding and Omics Research, Khon Kaen University, Khon Kaen 40002, Thailand; 3Animal Feed Quality Research Unit, Faculty of Veterinary Science, Mahasarakham University, Mahasarakham 44000, Thailand

**Keywords:** body weight, genetic parameter, hot-humid conditions, indigenous chicken

## Abstract

Heat stress is becoming a major problem because it limits growth in poultry production, especially in tropical areas. The development of genetic lines of Thai native chickens (TNC) which can tolerate the tropical climate with the least compromise on growth performance is therefore necessary. This research aims to analyze the appropriate growth curve function and to estimate the effect of heat stress on the genetic absolute growth rate (AGR) in TNC and Thai synthetic chickens (TSC). The data comprised 35,355 records for body weight from hatching to slaughtering weight of 7241 TNC and 10,220 records of 2022 TSC. The best-fitting growth curve was investigated from three nonlinear regression models (von Bertalanffy, Gompertz, and logistic) and used to analyze the individual AGR. In addition, a repeatability test-day model on the temperature-humidity index (*THI*) function was used to estimate the genetic parameters for heat stress. The Gompertz function produced the lowest mean squared error (MSE) and Akaike information criterion (AIC) and highest the pseudo-coefficient of determination (Pseudo-R^2^) in both chicken breeds. The growth rates in TSC were higher than TNC; the growth rates of males were greater than females, but the age at inflection point in females was lower than in males in both chicken breeds. The *THI* threshold started at 76. The heritability of the AGR was 0.23 and 0.18 in TNC and TSC, respectively. The additive variance and permanent environmental variance of the heat stress effect increased sharply after the *THI* of 76. The growth rate decreased more severely in TSC than TNC. In conclusion, the Gompertz function can be applied with the *THI* to evaluate genetic performance for heat tolerance and increase growth performance in slow-growing chicken.

## 1. Introduction

The local chicken breeds are significant for the rural economies of several countries [1] and are considered a genetic resource for use in the development of high-yielding breeds. The local breeds have been used as the foundation stock for breeding through a crossbreeding system with commercial breeds by exploiting heterosis. Also, the excellent characteristics of native chickens are their strengths, resistance to harsh environmental conditions, and poor rearing, without much loss in production. It has become essential to make farm management easier [2]. Unfortunately, many local breeds risk extinction (28.83% of local breeds are at risk) because of genetic erosion from government policy and programs, and climate change in terms of heat stress [3,4,5]. Therefore, conservation and sustainable development of animal genetic resources (AnGR) requires a broad focus on the many ‘adaptive’ breeds that survive well in the low external input agriculture typical of developing countries.

Heat stress profoundly decreases productivity and increases the mortality of poultry [4,5]. Thai native chickens (TNC) have evolved in tropical climates and are supposed to be well adapted to them; however, lower growth performance and egg production have been reported, regardless of whether they were raised in hot and humid conditions compared with other conditions [6,7]. Various methods have been proposed to mitigate the effect of heat stress, including housing and cooling systems, feed, and feed management [8,9,10], but there might not be an economically viable option for Thai farmers who raise TNC in a backyard farming system without temperature-controlled housing facilities. Genetic improvement through genetic selection for heat tolerance with the least compromise on growth performance within TNC seems to be a more permanent and sustainable option.

To mitigate the heat stress problems, the current focus of poultry genetic approaches is primarily on molecular genetics to select naked neck, frizzle, dwarf, and heat shock protein genes. This approach has been applied successfully in commercial broilers and layers [11,12,13,14]. However, there are few studies of these genes in native chickens, and the method may not be practical in large populations. In addition, molecular methods take a lot of time and money; therefore, the quantitative genetic approach in terms of associations between data records and climate data would be more efficient [15] and would help address the above-mentioned issues. In addition, there has never been a report on genetic selection for heat tolerance in chickens. This might be because all commercial chickens are raised under temperature-controlled housing facilities.

Growth traits are economically important and are easily susceptible to environmental changes [5]. To evaluate heat stress, selection for growth rates is desirable. The growth curve is widely used to design feeding programs suitable for age and to determine selection in breeding programs [16,17]. The growth curve is described by a non-linear function, namely sigmoidal. Common non-linear functions for describing growth patterns are Gompertz, logistic, von Bertalanffy, and Richards; the choice of function depends on the breed or population structure [18,19,20]. In the cited studies, however, the researchers did not include the effect of heat stress in the models. Recently, absolute growth rates (AGR) have been used to describe the growth of chickens throughout their maturation. The synergistic consideration of growth traits and heat tolerance using a quantitative genetic approach is interesting and unreported. In this study, we examine for the first time the appropriate models for describing the growth curve and the influence of heat stress on the AGR from growth curve parameters in purebred TNC and TSC. The Kai Shee breed (100% Thai native) represents the nucleus herd (TNC), while the Kaimook e-san1 breed (50% Thai native and 50% broiler) represents TSC, a cross between Kai Shee males and broiler females. The TSC breeding strategy aims to benefit from the complementary effect of both Thai native (good adaptability in harsh environments and good meat taste) and commercial broiler (high growth rate) chickens.

## 2. Materials and Methods

### 2.1. Animal Ethics 

Birds were handled and managed according to the Guidelines of Experimental Animal Care of the National Research Council of Thailand. The project was reviewed and approved by KKU–TRF contract number RDG5320010 and RDG5720005.

### 2.2. Animals and Management

The experiment was carried out in the experimental farm of the Network Center for Animal Breeding and Omics Research, Faculty of Agriculture, Khon Kaen University, Thailand, located in the Northeastern region of the country, from 2015 to 2020. The chickens used in this study were divided into two groups (1) Thai native breed (called Kai Shee; TNC), the grandparent stock (nucleus herd), and (2) Thai synthetic breed (called Kaimook e-san1; TSC), a commercial line. The base population is 30 males and 150 females (Ne = 100) Kai Shee chickens; this population is supported by the Thai Department of Livestock Development and were evaluated in this study. 

The chickens were raised in an open housing system under a pen with 1 m^2^ per eight chickens. At 20 weeks of age, both male and female chickens were raised in individual cages. They received an average of 12 h of natural light per day. The chicks were fed *ad libitum* a diet consisting of protein 19% energy 2900 kcal/kg from 0 to 4 weeks of age, followed by a diet containing protein 15% and energy 2900 kcal/kg from 4 to 20 weeks of age. 

For the breeding plan, the parents of the TNC G0 generation were Kai Shee; for TSC, a broiler was used as a cock and a Kai Shee was used as a hen. The F1 generations of both chicken breeds were selected and mated by using an inter se mating system. Each cock was mated to five hens (artificial insemination) by within-breed selection in a closed nucleus system, and then fertilized to produce eggs that hatch in sets. Three sets of the same parent pairs were bred, each incubated 1 week apart, followed by two more cycles of mating with alternating hens each round; each round produced three sets of offspring. In each generation there would be approximately 1500–1800 chicks, which are selected to produce 30 cocks and 150 hens to fertilize the next generation. The selection process was divided into two steps: (1) selection for true or morphological selection of the breed (Figure 1) and (2) genetic selection by using a selection index. Any chickens that do not match the breed characteristics at any stage of maturation are excluded from the flock. Each generation, replacement cocks and hens are also selected using a selection index to produce the next generation of offspring. For the genetic selection process, the cocks are selected with a focus on growth and breast circumference traits; individual animal genetics are assessed using a bivariate trait best linear unbiased prediction (BLUP) model from the resulting estimated breeding values (EBV) to analyze a selection index. Hens are placed in individual cages to assess egg production from the age at first egg to 300 days of age. Chickens are genetically assessed considering three traits, namely body weight, breast circumference at 16 weeks of age, and total egg production up to 300 days of age.

High-indexed male and female chickens are selected in each mating cycle (three hatching sets). In each cycle, approximately 10–15 cocks and 60–80 hens are selected. Therefore, after completing three cycles of selection, there will be at least 40 cocks and 240 hens for further testing. In the final testing at 32 weeks of age, cocks are tested for their ability to produce semen, and their semen quality is checked. Meanwhile, hens are assessed regarding whether they can lay eggs; any hens with low egg production and unable to produce eggs are eliminated. Top 30 cocks and 150 hens of high performance are selected after this procedure. The breeding plan and selection strategy is shown in Figure 2.

### 2.3. Climate Conditions

Climate data were obtained from the meteorological center closest to the chicken farm (3 km distance). The weather information included daily temperature and relative humidity recorded every 3 h, which were used to calculate the temperature-humidity index (*THI*) based on the formula [21]:*THI* = (1.8T + 32) − (0.55 − 0.0055RH)(1.8T − 26),
where T is the temperature in degrees Celsius and RH is the relative humidity as a percentage. The mean daily *THI* 4 weeks before each chicken weighing date was used to determine the heat stress threshold and assess genetic parameters.

The monthly average *THI* in Khon Kaen province, Thailand, over the past 5 years (between 2015 and 2020) has ranged from 72.60 to 81.36. The month with the highest *THI* is May, while the month with the lowest *THI* is December. These months correspond to the summer and winter in Thailand, respectively (Figure 3).

### 2.4. Data and Statistical Analysis

A total of 35,355 records for body weight from hatching to slaughtering weight (16 weeks of age) of 7241 TNC and 10,220 records of 2022 TSC were used in this study. Body weight for TNC and TSC were carried out in the same way, every 4 weeks. The growth curves for poultry generally have the following characteristics: an accelerated growth phase from hatching, a turning point in the growth curve where the growth rate is maximum, a phase when the growth rate slows down, and a threshold (asymptote) of mature weight [22]. The parameters of growth curves were estimated by using a nonlinear function—von Bertalanffy, Gompertz, and logistic—in the NLIN procedure of SAS [23]. The equations for the three growth curve models are given below.

$W_{\left(t\right)}=\alpha {\left(1-\beta e^{-\gamma t}\right)}^{3}$     for the von Bertalanffy function,$W_{\left(t\right)}=\alpha e{{(}^{-\beta e})}^{-\gamma t}$           for the Gompertz function, and$W_{\left(t\right)}=\alpha /(1+\beta e^{-\gamma t})$     for the logistic function.

In the above equations, $W_{\left(t\right)}$ is the corresponding body weight (grams) at time *t*, α is the asymptotic live body weight (grams), β is the the log-function for the proportion of the asymptotic mature weight to be gained after hatch (weeks), and γ is a constant scale that is proportional to the overall growth rate (weeks). 

The growth parameters were analyzed by using the Marquardt method NLIN procedure of SAS. The fit of the growth models (which is suitable for the non-linear regression model) was assessed by evaluating the mean squared error (MSE), the pseudo-coefficient of determination (Pseudo-R^2^) and Akaike information criterion (AIC) [24,25]. After that, parameters α,β, and γ were used to estimate the age at inflection point (IPA), the weight at inflection point (IPW), and the maximum growth rate (MGR) [26]. The equations used to estimate the IPA, IPW, and MGR are presented in Table 1. The best-fitting growth curve models (lowest MSE, highest R^2^ and lowest AIC) for different genders of both TNC and TSC were used to analyze the individual AGR and used as the observation value (y) to estimate the genetic parameters for heat stress. The AGR was calculated based on the first derivative from the adjusted function in relation to time (∂Y/∂t). In fact, the AGR represents the weight gained per time unit. The equation used to calculate the AGR of the three-growth curve model are shown in Table 1. The least squares mean for each chicken breed was plotted against age to obtain growth curve patterns.

### 2.5. Estimation of Genetic Parameters

For genetic analyses, a repeatability test-day model with the *THI* was used for both TNC and TSC. The *THI* included in the repeatability test-day model was set at various critical values or threshold points. Different thresholds of the *THI*, from 72 to 81, were tested in the model. The best model was considered to come from the lowest minus twice the logarithm of the likelihood (−2logL), and Akaike information criterion (AIC). Heritability and genetic correlation between the intercept (the intercept of the random additive effect indicates the animal’s ability to produce body weight in thermoneutral conditions) and slope (the slope of the additive genetic effect describes the animal’s sensitivity to heat stress; it represents the change in body weight per *THI* unit increase above a given threshold). The *THI* function was created to estimate the decline in growth performance under heat stress conditions. The *THI* function used in the equation was defined as follows:$$f\left(THI\right)=\{\begin{array}{cc} 0, & THI\leq THI_{threshold}\;\left(no\;heat\;stress\right)\\ THI-THI_{threshold}, & THI>THI_{threshold}\;\left(heat\;stress\right) \end{array}.$$

The repeatability test-day model and variance covariance structure matrix were:$$y_{ijkl}=CG_{i}+SEX_{j}\left[f\left(THI\right)\right]+a_{0k}+a_{1k}\left[f\left(THI\right)\right]+p_{0k}+p_{1k}[f\left(THI\right)+e_{ijkl}$$
$$Var\left[\begin{array}{c} a_{0}\\ a_{1}\\ p_{0}\\ p_{1}\\ e \end{array}\right]=\left[\begin{array}{ccccc} A\sigma_{a0}^{2} & A\sigma_{a01} & 0 & 0 & 0\\ A\sigma_{01} & A\sigma_{a1}^{2} & 0 & 0 & 0\\ 0 & 0 & I\sigma_{p0}^{2} & I\sigma_{p01} & 0\\ 0 & 0 & I\sigma_{p01} & I\sigma_{p1}^{2} & 0\\ 0 & 0 & 0 & 0 & I\sigma_{e}^{2} \end{array}\right],$$
where $y_{ijkl}$ is the observation value of the AGR of chicken *l*; $CG_{i}$ is the fixed effect of the contemporary group (hatch and generation); *i*, $SEX_{j}$ is the fixed effect of sex *j*, which is the slope (regression coefficient) of change in the AGR per unit of change according to the sex of the chicken; $a_{0k}$ and $\sigma_{a0}^{2}$ are the random additive genetic effects without consideration of heat tolerance (intercept); $a_{1k}$ and $\sigma_{a1}^{2}$ are the random additive genetic effects for heat tolerance (slope); $p_{0k}$ and $\sigma_{p0}^{2}$ are the random permanent environmental effects without consideration of heat stress (intercept); $p_{1k}$ and $\sigma_{p1}^{2}$ are the random permanent environmental effects of heat tolerance (slope); $e_{ijkl}\;$and$\;\sigma_{e}^{2}$ are the random residual effect; f(THI) is a function of the *THI*; A is the numerator relationship matrix; and I is an identity matrix.

Variance components were estimated with the average information restricted maximum likelihood (AI-REML) method by using the BLUPF90 Chicken PAK v. 2.5 program [27]. The heritability of the AGR under hot-humid climates and genetic correlation were estimated as reported by Ravagnolo and Misztal [28]. Heritability and permanent environmental variance were calculated as:$$h^{2}=\frac{\sigma_{a0}^{2}+\sigma_{a1}^{2}+2\sigma_{a01}}{\sigma_{a0}^{2}+\sigma_{a1}^{2}+2\sigma_{a01}+\sigma_{p0}^{2}+\sigma_{p1}^{2}+2\sigma_{p01}+\sigma_{e}^{2}}\;,pe^{2}=\frac{\sigma_{p0}^{2}+\sigma_{p1}^{2}+2\sigma_{p01}}{\sigma_{a0}^{2}+\sigma_{a1}^{2}+2\sigma_{a01}+\sigma_{p0}^{2}+\sigma_{p1}^{2}+2\sigma_{p01}+\sigma_{e}^{2}}.$$

Genetic correlations $\left(r_{g}\right)$ between the intercept and slope of the additive genetic effects and correlations between the intercept and slope of the permanent environmental effects $\left(r_{p}\right)$ were calculated as:$$r_{g}=\frac{COV\sigma_{a0,a1}}{\sqrt{\sigma_{a0}^{2}*\sigma_{a1}^{2}}}\;,r_{p}=\frac{COV\sigma_{p0,p1}}{\sqrt{\sigma_{p0}^{2}*\sigma_{p1}^{2}}}$$

## 3. Results

In the present study, we first analyzed the appropriate growth curve function and estimated the effect of heat stress on the genetic AGR traits in TNC and TSC. The Gompertz function was found to be suitable to estimate the shape of the growth curve in both TNC and TSC. The growth rate in terms of the IPW and MGR in males was greater than in females, but the IPA in females was younger than in males. The threshold point of heat stress was found at the *THI* of 76. The heritability of the AGR traits was moderate, but the additive variance and permanent environmental variance of the heat stress effect increased sharply after the *THI* of 76. The growth rate decrease was more pronounced in TSC compared with TNC.

### 3.1. Fitting Model and Growth Curve Parameters

The parameters for the growth curve model are shown in Table 2. The mature weight (α) and growth rate (γ) for male chickens were higher than for female chickens for both Thai native and synthetic lines. Three growth curve functions were used to fit the growth curve of TNC and TSC from hatching to 16 weeks of age (Table 2). For both chicken breeds, the model based on the Gompertz function presented the lowest MSE and AIC and highest Pseudo-R^2^. The IPA, IPW, and MGR in the Gompertz function were higher in males compared with females of both TNC and TSC. Each of these parameters was higher in TSC compared with TNC. The IPA in male TNC was slower than in female TNC by about 0.3 weeks; this difference was about 0.1 weeks in TSC. The difference in the IPW between males and females was 157.8 g in TNC and 171.4 g in TSC.

### 3.2. The Characteristics of the Growth Curve Based on the Gompertz Function

The pattern of the growth curve based on the Gompertz function is shown in Figure 4. The AGR of male chickens of both breeds was higher compared with females based on estimates with the Gompertz function (Figure 4a,b). The AGR increased from 56.2 to 99.5 g/day between 4 and 12 weeks of age and decreased by 5.9% from 12 to 16 weeks of age in TNC. The estimated maximum AGR of our studies occurred at 9 weeks of age in TNC (Figure 4c) and 7 weeks of age in TSC (Figure 4d). These data are consistent with the description of the IPA based on the Gompertz function (Table 2). 

### 3.3. Estimated Genetic Parameters

The indicated threshold point was found at the *THI* of 76 based on the lowest −2logL and AIC. The variance components and genetic parameters of the AGR using the Gompertz function in TNC and TSC at the *THI* of 76 are shown in Table 3 and changes of variances and heritability after *THI* threshold are shown in Figure 5. The heritability ($h^{2}$) of the AGR traits was 0.23 and 0.18 in TNC and TSC, respectively. The permanent environmental variance ($pe^{2}$) was about three times that of $h^{2}$. The genetic correlation between the AGR and the heat stress effect was −0.77 and −0.82 in TNC and TSC, respectively. The rate of AGR decline in males (−10.10 and −20.53 g per *THI* unit increase for TNC and TSC, respectively) was greater compared with females (−6.10 and −11.35 g per *THI* unit increase for TNC and TSC, respectively). Of note, the decline was about two times higher in TSC compared with TNC.

## 4. Discussion

Heat stress is becoming a major problem because it reduces the growth of poultry, especially in tropical areas [29,30]. Therefore, it is necessary to develop genetic lines of native chickens that can tolerate the tropical climate with the least compromise on growth performance [31]. The results of this study can be applied in farm management planning, when chickens are experiencing heat stress when *THI* value is higher than 76. This will help farmers prepare to cope and find a way to cool the chickens. At the same time, the problem of heat stress is likely to continue to increase. Therefore, chicken breeding programs should be considered for promoting the genetic potential of production and to adapting to harsh weather conditions. The new finding of this study is a new genetic model that can be used quickly and easily in a large data set.

The Gompertz function produced the growth curve with the lowest MSE and AIC and highest Pseudo-R^2^ compared with the other functions. These data suggest that the Gompertz function is suitable to estimate the shape of the growth curve in both TNC and TSC populations. This result is consistent with several reports in which the Gompertz function has been used to describe slow-growth chickens [32,33]. TNC have a slower growth rate and reach a smaller size at maturity than TSC, suggesting that they do not need high energy concentrations in their feed [34]. Meanwhile, the growth rate in terms of the IPW and MGR in males was greater than in females, but the IPA in females was younger than in males. These data indicate that females reach maturity earlier and have a lower weight at maturity. We compared the IPA in TNC with other native chickens. We found that TNC have better genetic potential than other native chickens because the growth rate in terms of the IPA in TNC (9.4 weeks) is lower than native chickens from Italy, Ghana, and China (10.1–13.9 weeks) [26,35,36,37]. The IPA of TSC in our study was acceptable (just about 1–2 weeks later than commercial broilers). Hence, both TNC and TSC have potential as a chicken suitable for sustainable models. The many benefits derived from rearing native chickens include food sources of protein with high nutritional value, savings, and income generation for farmers and help drive a rural circular economy. Apart from being nutritious, native chicken products are inexpensive without social or religious customs prohibiting consumption. It is essential to acknowledge that native chickens play a significant role in rural communities in developing countries. The development and promotion of native chicken may be the answer to the perpetual food insecurity, poverty alleviation, and hunger reduction for the betterment of the rural populace.

We evaluated the growth curves and found that the body weight is increased dramatically starting at 4 weeks to either 16 weeks in TSC or 20 weeks in TNC. The peak of body weight was at 9.3 and 7.1 weeks of age in TNC and TSC, respectively. Hence, body weight data from weeks 7 to 9 are enough to evaluate genetic effects. This is an advantage because it means spending less time raising chickens until they reach maturity or slaughtering weight [38], an endeavor that is time-consuming for genetic evaluation. 

We found that the heritability estimates of the AGR were moderate (Table 3). The values were similar for both chicken breeds at the *THI* threshold of 76, namely 0.23 and 0.18 for TNC and TSC, respectively. These data are similar to findings in Korean native chickens as reported heritability estimates of growth curve parameters (0.10–0.25) [39]. In general, the heritability estimates for growth traits throughout life vary in native chickens and depend on population [38,39]. Previous studies in poultry have reported decreased heritability estimates of weight gain with increasing age [40,41]. The moderate heritability suggests that by using a conventional method with the AGR, it is possible to evaluate genetic effects in chicken. 

The permanent environmental variance was about three times the heritability, reflecting that the environmental factors are more important than genetic factors. The reduction in animal productivity is a consequence of changes in physiology and behavior in response to heat stress, which has been reported in several studies [41,42,43,44]. Therefore, intensive management is needed to promote better growth performance including nutritional strategies such as restricted feeding and maintaining feed intake, choice feeding from different feed ingredients, rich in protein or in energy, electrolytic and water balance, and supplementing micronutrients [13,45]. Meanwhile the genetic correlation estimates between regular conditions and heat stress were very high, suggesting that chickens with a high growth rate have low heat tolerance. Hence, selecting chickens for a high growth rate would lead to chickens with greater susceptibility to heat stress [46,47]. Although the genetic correlation is very large, the combined selection of growth rate and heat tolerance using the breeding value index is possible [28,48].

After the chickens reach heat stress at the *THI* of 76, the growth rate is decreased. This finding suggests that heat stress negatively affects the growth ability. The growth rate decrease in terms of the average rate of decline of the AGR trait was greater in TSC than TNC (−15.94 and −8.36 g/*THI* unit increase, respectively), with a greater effect in male compared with female chickens. These different heat tolerances might be because the Kai Shee breed is derived from native chickens and is more tolerant to heat. Hence, the *THI* threshold before heat stress affects the growth rate is higher in TNC than TSC, similar to what has been reported previously [6,7]. Indeed, those studies indicate higher heat tolerance in TNC than in commercial broiler strains when reared in hot temperatures.

The additive variance and permanent environmental variance of the effect of heat stress increased sharply after the *THI* of 76. These data indicate that the chickens express genetic factors that allow them to cope with heat stress. There was decreased heritability of these factors at the higher *THI*, and thus animal selection using genetics is not precise.

## 5. Conclusions

This is the first report on the genetic evaluation of heat stress in native chickens. The Gompertz function is suitable to estimate the shape of the growth curve in both purebred Thai native and Thai synthetic chickens raised in the open housing system. Both purebred Thai native and Thai synthetic chickens can maintain growth performance up to *THI* of 76 despite a lack of active management practices to alleviate such stress, after which heat stress occurs. In addition, the genetic model that combines the growth curve model and the *THI* function will help in genetic selection for chickens with both high growth rate and good adaptation to harsh environments. In a further study, we can apply this knowledge to other economic traits or even in multiple traits in poultry.

## Figures and Tables

**Figure 1 vetsci-08-00297-f001:**
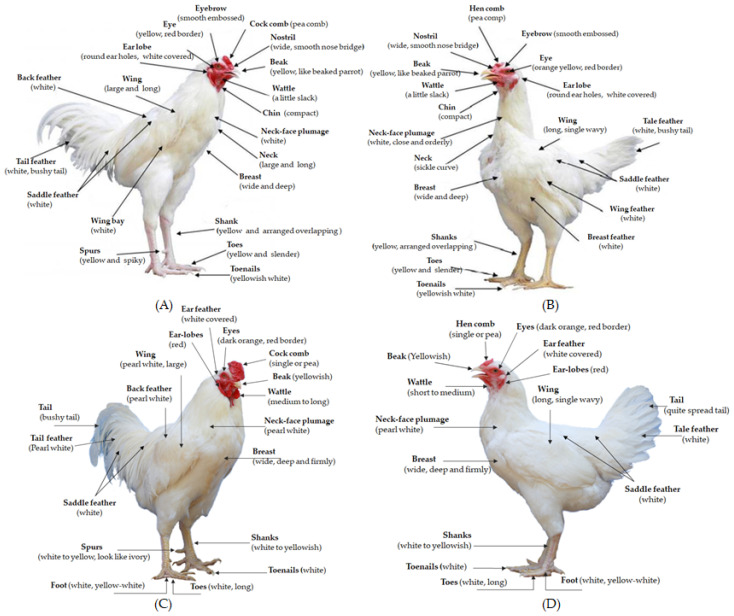
Morphology of male (**A**) and female (**B**) purebred Thai native chickens (Kai Shee) and male (**C**) and female (**D**) Thai synthetic chickens (Kaimook e-san1).

**Figure 2 vetsci-08-00297-f002:**
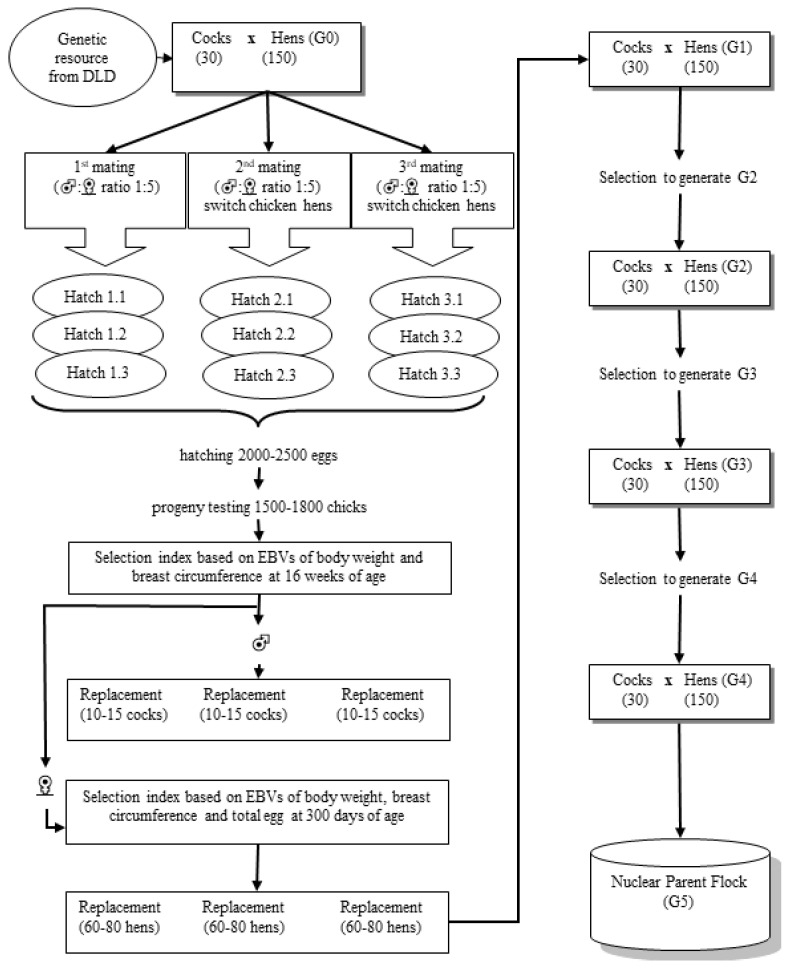
Breeding plan and selection strategy of purebred Thai native and Thai synthetic chickens. Abbreviations: DLD, Thai Department of Livestock Development; EBV, estimated breeding values.

**Figure 3 vetsci-08-00297-f003:**
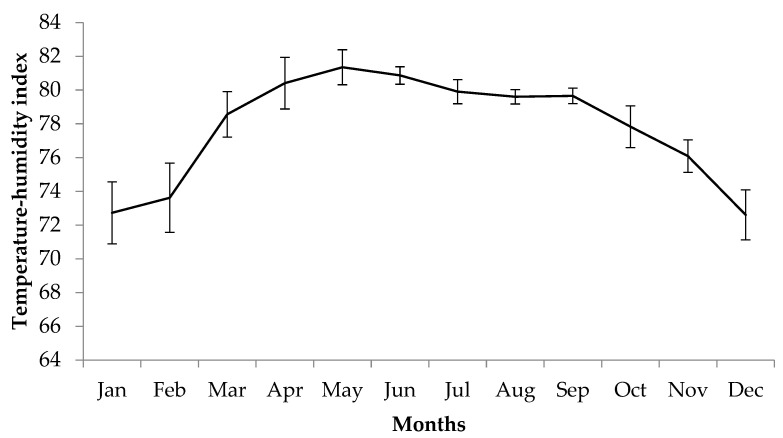
Average temperature-humidity index in Khon Kean province, Thailand, during 2015–2020.

**Figure 4 vetsci-08-00297-f004:**
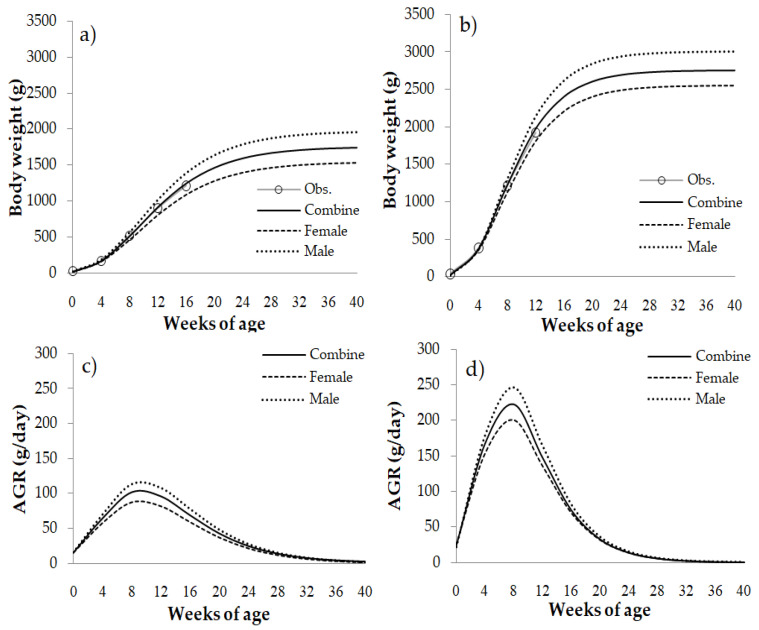
Characteristics of the estimated growth curves (body weight and the absolute growth rate [AGR]) using the Gompertz function in purebred Thai native chickens (**a**,**c**) and Thai synthetic chickens (**b**,**d**).

**Figure 5 vetsci-08-00297-f005:**
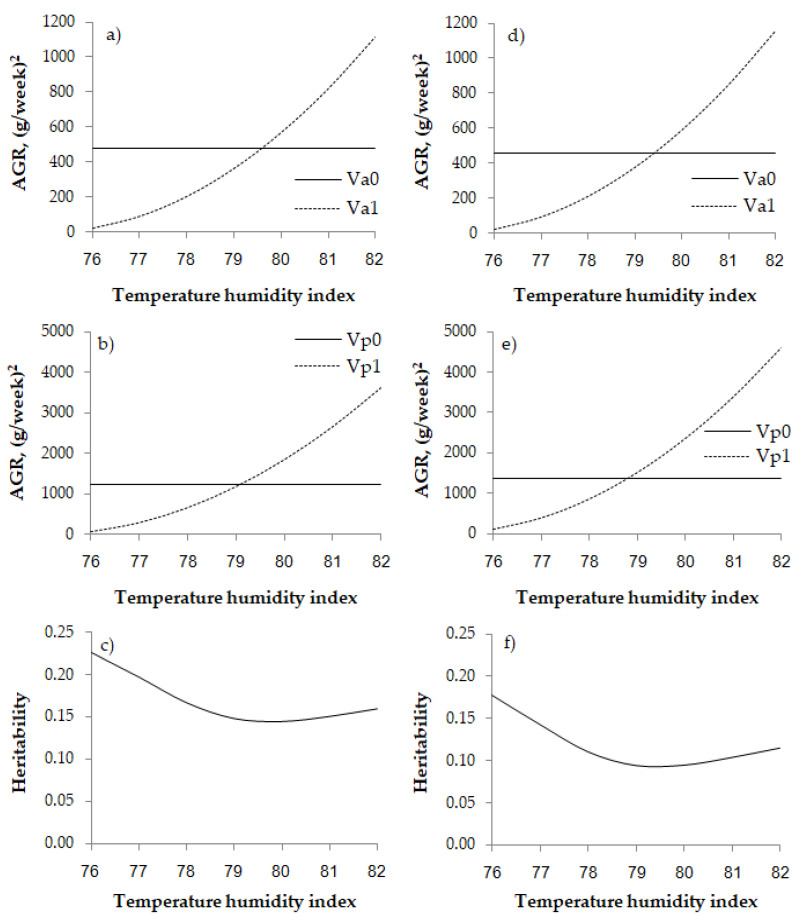
Estimates of additive genetic variance without consideration of heat tolerance (intercept, Va0, solid line), additive genetic variance for heat tolerance (slope, Va1, dashed line) (**a**,**d**), permanent environmental variance without consideration of heat stress (intercept, Vp0, solid line), permanent environmental variance of heat tolerance (slope, Vp1, dashed line) (**b**,**e**), and heritability (**c**,**f**) using the Gompertz function in purebred Thai native chickens (all left figures) and Thai synthetic chickens (all right figures).

**Table 1 vetsci-08-00297-t001:** The growth functions used to estimate the growth curve characteristics.

Functions	von Bertalanffy	Gompertz	Logistic
Characteristics			
Age at inflection point (IPA)	(ln3β)/γ	ln(β)/γ	ln(β)/γ
Weight at inflection point (IPW)	8α/27,	α/e,	α/2
Maximum growth rate (MGR)	3γIPW/2	γIPW	γIPW/2
Absolute growth rate (AGR)	$3\alpha \gamma {\left(1-\beta e^{-\gamma t}\right)}^{2}\beta e^{-\gamma t}$	$\alpha \beta \gamma e^{\left(-\beta e^{-\gamma t}\right)}e^{-\gamma t}$	$\alpha \beta \gamma e^{\left(-\gamma t\right)}/{\left(1+\beta e^{-\gamma t}\right)}^{2}$

**Table 2 vetsci-08-00297-t002:** Nonlinear regression functions used to fit the growth curves of purebred Thai native and Thai synthetic chickens.

Functions	Sex	Parameters			Model fit			Growth Inflection		Maximum Growth Rate
		α	β	γ	MSE	Pseudo-R^2^	AIC	IPA (wk)	IPW (g)	MGR (g/wk)
	Thai native ( Kai Shee)									
von Bertalanffy	Male	2495.1	0.88	0.10	54,871	0.94	188,357	9.5	739.3	112.1
	Female	2004.9	0.82	0.09	17,621	0.95	176,916	9.4	594.0	84.8
	Combine	2252.4	0.85	0.10	38,083	0.94	376,789	9.5	667.4	98.9
Gompertz	Male	1979.2	4.50	0.16	**54,722**	**0.97**	**188,310**	**9.4**	**728.1**	**116.4**
	Female	1550.2	4.13	0.15	**17,519**	**0.96**	**176,810**	**9.1**	**570.3**	**88.5**
	Combine	1760.6	4.39	0.16	**37,948**	**0.96**	**376,664**	**9.3**	**647.7**	**103.0**
logistic	Male	1515.7	27.01	0.34	55,044	0.92	188,412	9.7	757.9	128.7
	Female	1185.7	22.54	0.33	17,703	0.96	177,000	9.4	592.9	97.8
	Combine	1340.4	26.24	0.34	38,206	0.94	376,904	9.6	670.2	114.4
	Thai synthetic (Kaimook e-san1)									
von Bertalanffy	Male	3524.5	0.98	0.16	47,033	0.94	54,485	6.9	1044.3	244.4
	Female	3029.1	0.92	0.15	33,891	0.96	53,785	6.9	897.5	198.6
	Combine	3205.9	0.96	0.16	47,851	0.95	110,132	6.8	949.9	221.4
Gompertz	Male	3012.5	5.07	0.23	**46,195**	**0.97**	**54,426**	**7.2**	**1108.2**	**250.1**
	Female	2554.6	4.67	0.22	**33,243**	**0.97**	**53,716**	**7.1**	**939.8**	**203.7**
	Combine	2752.5	4.90	0.22	**47,047**	**0.96**	**110,024**	**7.1**	**1012.6**	**226.4**
logistic	Male	2345.5	33.02	0.47	46,256	0.92	54,389	7.5	1172.8	274.5
	Female	1982.5	28.09	0.45	33,412	0.94	53,680	7.4	991.3	223.9
	Combine	2150.4	30.69	0.46	47,311	0.93	109,951	7.4	1075.2	248.7

α, asymptotic live body weight (grams); β, the log-function for the proportion of the asymptotic mature weight to be gained after hatching (weeks); γ, a constant scale that is proportional to the overall growth rate; MSE, the mean squared error; Pseudo-R^2^, the pseudo-coefficient of determination; AIC, Akaike information criterion; IPA, age at inflection point; IPW, weight at inflection point; MGR, maximum growth rate.

**Table 3 vetsci-08-00297-t003:** Variance components and genetic parameters of the absolute growth rate (AGR) using the Gompertz function in purebred Thai native and Thai synthetic chickens at the temperature-humidity index (*THI*) of 76.

Parameter Estimates	Thai Native Breed	Thai Synthetic Breed
$\sigma_{a0}^{2}$	478.58	455.20
$\sigma_{a1}^{2}$	22.70	23.52
$\sigma_{a01}$	−80.56	−84.38
$\sigma_{p0}^{2}$	1223.95	1352.30
$\sigma_{p1}^{2}$	74.03	94.10
$\sigma_{p01}$	−190.82	−222.44
$\sigma_{e}^{2}$	249.660	428.32
$h^{2}$ (SE)	0.23 ± 0.01	0.18 ± 0.03
$pe^{2}$ (SE)	0.61 ± 0.01	0.58 ± 0.02
$r_{g}$	−0.77	−0.82
$r_{p}$	−0.63	−0.62
Rate of decline of AGR		
Male (g/*THI*)	−10.10	−20.53
Female (g/*THI*)	−6.61	−11.35
Average (g/*THI*)	−8.36	−15.94

$\sigma_{a0}^{2}$, additive genetic variance without consideration of heat tolerance (intercept); $\sigma_{a1}^{2}$, additive genetic variance for heat tolerance (slope); $\sigma_{a01}$, additive genetic covariance between intercept and slope; $\sigma_{p0}^{2}$, permanent environmental variance without consideration of heat stress (intercept); $\sigma_{p1}^{2}$, permanent environmental variance of heat tolerance (slope); $\sigma_{p01}$, permanent environmental covariance between intercept and slope; $\;\sigma_{e}^{2}$, residual variance; $h^{2}$, heritability; $pe^{2}$, permanent environmental variance; $r_{g}$, genetic correlations between intercept and slope; and $r_{p}$, correlations between the intercept and slope of the permanent environmental effects.

## Data Availability

The data are available upon request of the corresponding author.

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
