# Peer review of "Genetic Effect and Growth Curve Parameter Estimation under Heat Stress in Slow-Growing Thai Native Chickens"

_vetsci, 2021, doi:10.3390/vetsci8120297_

Round 1

Reviewer 1 Report

Dear authors,

First at all, I want to thank for the possibility to review this manuscript, which contains interesting information for the scientific community since it provides important insights about the approaches in some current issues such as global warming and the ecological problems we are facing and the development of sustainable genotypes adapted to alternative production systems.

Nevertheless, I have some issues which I would like to be clarified:

Paragraph formatting, font type and indentation corrections must be made in the full text (including tables and images).

Line 38: it is not appropriate to use the same words or expressions in the title and in the keywords. Please rewrite the expression “native chicken”.

Line 40: It would be interesting if more information was provided on the benefits of developing farms using local breeds and why animal genetic resources should be conserved.

Line 74: Is Chee breed considered by the DAD-IS (FAO)? If so, indicate it and say the name that this breed receive on this platform to facilitate the reader’s search.

Line 115: why did you select that range (age at the first egg to 300 days of age) to measure egg production? Do you think that body weight and breast circumference should have been measured in hens? Please clarify it.

Lines 120-124: you explain in the test that 110 of 240 hens were eliminated when were assessed regarding whether they can lay eggs. If so, I think there would be a reproductive problem in the study population. Could you explain if this is really the case or is there a problem in the transfer of information to the reader?

Line 127: I think that the font size should be increased for greater readability. In addition, the letters (A, B, C and D) can only be seen partially.

Line 149: Weighings were carried out every 4 weeks in both genotypes? Please clarify it.

Lines 165-166: R2 in linear regression models determine the quality of the fit of the used model. However, in non-linear regression models, R2 could overestimate results. So, pseudo-R2 must be used to correct for the potential inflation of variance explanatory ability and to avoid the use of data that may distort the results (González Ariza et al., 2021).

  • González Ariza, et al. "Characterisation of biological growth curves of different varieties of an endangered native hen breed kept under free range conditions." Italian Journal of Animal Science 20.1 (2021): 806-813.

In addition, both R2 and MSE are goodness-of-fit criteria, but in this case, flexibility criteria (AIC, AICc and BIC) can quantify the explanatory and predictive ability of the models tested. It would be interesting if these criteria were included in order to provide greater quality to this study (Arando et al., 2021).

  • Arando, A., et al. "Comparison of non-linear models to describe the growth in the Andalusian turkey breed." Italian Journal of Animal Science 20.1 (2021): 1156-1167.

Line 157, line 169 and line 178: I think that the use of one or two tables where equations, IPA, IPW, MGR, and AGR are presented is necessary to clarify the exposition of the used methods.

Line 261: I do not understand the use of superscripts 1 and 2 in the table header. Please clarify it.

Line 322: Figure 5 is not clear. it must be joined and displayed on a single page.

Line 326: in the discussion, you state the results found in the two genotypes under study, but you do not seek an applicability of these results on farms. It is necessary to show what is the scientific advance that you have found in this study, which can be very interesting from a practical point of view. What should, therefore, be the strategies they should follow

Lines 331-338: this part of the paragraph is more suitable for the results section than for the discussion section. Here you are explaining your results, but you should also discuss these results in greater depth.

Line 339: What are the growth characteristics of these two genotypes so that the Gompertz model is the one that best fits them?

Line 348-353: you talk about suitability of chickens of different breeds for meat production, but you do not explain in which production systems in which these breeds are developed. Local breeds cannot be seen from an agribusiness point of view. We must look for what is the benefit that these genotypes in sustainable models.

Line 357-359: add a bibliographic reference. You are talking about a market weight, but rather a slaughtering weight should be talked about, since these results are not based on results or consumer preferences for this type of product.

Line 362-363: Please describe what are the findings made by the authors in Korean native chicken.

Line 371-377: add a bibliographic reference.

Line 379-380: You have not studied the feed intake, so you cannot make this statement.

Line 393: conclusions must be more consistent. Please explain what is the scientific development that this study can bring.

Author Response

Response to Reviewer 1 Comments

Point 1: Paragraph formatting, font type and indentation corrections must be made in the full text (including tables and images).

Response 1: We have edited the paragraph formatting, font type and indentation corrections must be made in the full text (including tables and images) as your suggestion. please see the revised manuscript.

Point 2: Line 38: it is not appropriate to use the same words or expressions in the title and in the keywords. Please rewrite the expression “native chicken”.

Response 2: We have changed the keyword from "native chicken" to "indigenous chicken". Please see line 31.

Point 3: Line 40: It would be interesting if more information was provided on the benefits of developing farms using local breeds and why animal genetic resources should be conserved.

Response 3: We added more information about the benefits of developing farms using local breeds and the important of animal genetic resources. Please see lines 34-45.

“The local chicken breeds are significant for the rural economies of several countries (Ariza et al., 2021a) and are considered a genetic resource for use in the development of high-yielding breeds. The local breeds have been used as the foundation stock for breeding through a crossbreeding system with commercial breeds by exploiting heterosis. Also, the excellent characteristic of native chickens is their strength, resistance to harsh environmental conditions, and poor rearing without much loss in production. It has become essential to make farm management easier (Padhi, 2016). Unfortunately, many local breeds risk extinction (28.83% of local breeds at risk) because of genetic erosion from government policy/programs and climate change (FAO, 2021). Therefore, conservation and sustainable development of animal genetic resources (AnGR) requires a broad focus on the many ‘adaptive’ breeds that survive well in the low external input agriculture typical of developing countries.”

Point 4: Line 74: Is Chee breed considered by the DAD-IS (FAO)? If so, indicate it and say the name that this breed receive on this platform to facilitate the reader’s search.

Response 4: Yes, Thai native chicken named “Kai Shee” exists in the Domestic Animal Diversity Information System (DAD-IS) database. We have adjusted the spelling of the name to match the DAD-IS. Please see in Manuscript.

Point 5: Line 115: why did you select that range (age at the first egg to 300 days of age) to measure egg production? Do you think that body weight and breast circumference should have been measured in hens? Please clarify it.

Response 5: The days during age at first egg to 300 days of age are a normal range for genetic evaluation and selection in laying hens covering the beginning, peak, and decline phases of egg production. About body weight and breast circumference traits in hens, although both traits are not the primary traits for genetic improvement in chicken hens, however considering only egg production trait may lead to a genetic decrease in growth traits due to egg production and growth traits have a negative genetic correlation, therefore, in the breeding program of chicken hens it is necessary to consider both egg production and growth traits simultaneously.

Point 6: Lines 120-124: you explain in the test that 110 of 240 hens were eliminated when were assessed regarding whether they can lay eggs. If so, I think there would be a reproductive problem in the study population. Could you explain if this is really the case or is there a problem in the transfer of information to the reader?

Response 6: We apologize for the misunderstanding; we have rewritten the sentence as follows “any hens with low egg production and unable to produce eggs are eliminated. Top 30 cocks and 150 hens of high performance are selected after this procedure.” Moreover, from our data in this study, there are 5% of hens that unable to produce egg. Please see lines 131-133.

Point 7: Line 127: I think that the font size should be increased for greater readability. In addition, the letters (A, B, C and D) can only be seen partially.

Response 7: We have adjusted according to your suggestion. Please see Figure 1.

Point 8: Line 149: Weighings were carried out every 4 weeks in both genotypes? Please clarify it.

Response 8: Yes, in this study we weighed every 4 weeks in both genotypes. We have rewritten the sentence as follows “

“A total of 35,355 records for body weight from hatching to slaughtering weight (16 weeks of age) of 7,241 TNC and 10,220 records of 2,022 TSC were used in this study. Body weight for TNC and TSC were carried out in the same way, every 4 weeks.”. Please see lines 158-160.

Point 9: Lines 165-166: Rin linear regression models determine the quality of the fit of the used model. However, in non-linear regression models, Rcould overestimate results. So, pseudo-R2 must be used to correct for the potential inflation of variance explanatory ability and to avoid the use of data that may distort the results (González Ariza et al., 2021).

Response 9: Thank you for your suggestion. We have recalculated and used pseudo-R2 instead of R2 and added more AIC values including citation references (González Ariza et al., 2021b and Arando et al., 2021). Moreover, we have already described in the data and statistical analysis section, Table 2 as well as the results section. Please see revised MS.

Point 10: In addition, both Rand MSE are goodness-of-fit criteria, but in this case, flexibility criteria (AIC, AICc and BIC) can quantify the explanatory and predictive ability of the models tested. It would be interesting if these criteria were included in order to provide greater quality to this study (Arando et al., 2021).

Response 10: Thank you for your suggestion. We already added more AIC in this study. Please see Table 2.  

Point 11: Line 157, line 169 and line 178: I think that the use of one or two tables where equations, IPA, IPW, MGR, and AGR are presented is necessary to clarify the exposition of the used methods.

Response11: We have created a new table to describe the IPA, IPW, MGR, and AGR equations. Please see Table 1.

Point 12: Line 261: I do not understand the use of superscripts 1 and 2 in the table header. Please clarify it.

Response12: We apologize for the misunderstanding; we have deleted superscripts 1 and 2 in the table header. Please see Table 3. 

Point 13: Line 322: Figure 5 is not clear. it must be joined and displayed on a single page.

Response13: We have modified the Figure 5 and displayed on a single page. Moreover, we have rewritten the sentence for clarify. Please see Figure 5.

Point 14: Line 326: in the discussion, you state the results found in the two genotypes under study, but you do not seek an applicability of these results on farms. It is necessary to show what is the scientific advance that you have found in this study, which can be very interesting from a practical point of view. What should, therefore, be the strategies they should follow

Response14: Thank you for your suggestion. We have rewritten the sentences as follow.

“In the present study, we can be used in farm management planning, especially when the THI is higher than 76, which is a warning to farmers that their chickens are experiencing heat stress. At the same time, the problem of heat stress is likely to continue to increase. Therefore chicken breeding program would be considered in both the genetic potential of production and can adapt well in harsh weather conditions. The new finding of this study is a new genetic model that can be used quickly and easily in a large data set.” Please see lines 339-345.

Point 15: Lines 331-338: this part of the paragraph is more suitable for the results section than for the discussion section. Here you are explaining your results, but you should also discuss these results in greater depth.

Response14: We have moved the sentences into result section. Also we have rewritten sentences. Please see in 339-345.

Point 16: Line 339: What are the growth characteristics of these two genotypes so that the Gompertz model is the one that best fits them?

Response16: In theory, the Gompertz model’s essential characteristic is its ability to exhibit exponential growth rates in several researches such as body weight in livestock animals including chickens (Dottavio et al., 2007: Dourado et al., 2009; EleroÄŸlu et al., 2014; Mignon-Grasteau et al., 1999), and also the Gompertz model was used to study the growth rate of tumor and cancer in medicine. In other words, the Gompertz model embodies the fact that performance growth rates increase or decrease as a function of time. The Gompertz equation had been initially constructed for actuarial analysis but later came into use as a growth curve.

Citation references:

  1. Dottavio, A.M.; Álvarez, M.; Canet, Z.E.; Font, M.T.; Di Masso, R.J. Growth pattern of experimental hybrids for free range broiler production. Arg. Prod. Anim. 2007, 27, 75–82.
  2. Dourado, L.R.B.; Sakomura, N.K.; Nascimento, D.C.N.; Dorigam, J.C.; Marcato, S.M.; Fernandes, J.B.K. Growth and performance of naked neck broiler reared in free–range system. Ciênc. Agrotec. 2009, 33, 875–881.
  3. EleroÄŸlu, H.; Yıldırım, A.; SekeroÄŸlu, A.; Çoksöyler, F.N.; Duman, M. Comparison of growth curves by growth models in slow–growing chicken genotypes raised the organic system. J. Agric. Biol. 2014, 16, 529–535.
  4. Mignon-Grasteau, S.; Beaumont, C.; Le Biham–Duval E.; Poivey, J.P.; De Rochembeau, H.; Ricard, F. H. Genetic parameters of growth curve parameters in male and female chickens. Poult. Sci. 1999, 40, 44–51.

Point 17: Line 348-353: you talk about suitability of chickens of different breeds for meat production, but you do not explain in which production systems in which these breeds are developed. Local breeds cannot be seen from an agribusiness point of view. We must look for what is the benefit that these genotypes in sustainable models.

Response17: Thank you for your suggestion and we agree with you. Therefore we have been rewritten as the sentences below.

“Hence, both TNC and TSC have potential as a chicken suitable for sustainable models. The many benefits derived from rearing native chickens include food sources of protein with high nutritional value, savings, and income generation for farmers and help drive a rural circular economy. Apart from being nutritious, native chicken’s products are inexpensive without social or religious customs prohibiting consumption. It is essential to acknowledge that native chickens play a significant role in rural communities in developing countries. The development and promotion of native chicken may be the answer to the perpetual food insecurity, poverty alleviation, and hunger reduction for the betterment of the rural populace.” Please see lines 359-368.

Point 18: Line 357-359: add a bibliographic reference. You are talking about a market weight, but rather a slaughtering weight should be talked about, since these results are not based on results or consumer preferences for this type of product.

Response17: We agree with you and change "market weight" to "slaughtering weight" and added a bibliographic reference. Please see line 374.

Citation reference:

  1. Tongsiri, S., Jeyaruban, Gilbert & Hermesch, Susanne & Werf, Julius & Li, Li & Chormai, Teerachai. (2019). Genetic parameters and inbreeding effects for production traits of Thai native chickens. Asian-Australasian Journal of Animal Sciences. 32. 10.5713/ajas.18.0690.

Point 19: Line 362-363: Please describe what are the findings made by the authors in Korean native chicken.

Response 19: These data are similar to findings in Korean native chickens as reported heritability estimates of growth curve parameters (0.10-0.25). Please see lines 377-378.

Point 20: Line 371-377: add a bibliographic reference.

Response 20: We have already added content-related bibliographic references to this paragraph, which include: Settar et al., 1999; Ravagnolo and Misztal, 2000; Al-Batshan, 2002; Usala et al., 2021. Please see lines391-369 and in reference section.

Point 21: Line 379-380: You have not studied the feed intake, so you cannot make this statement.

Response 21: We have removed this sentence from the manuscript.

Point 22: Line 393: conclusions must be more consistent. Please explain what is the scientific development that this study can bring.

Response 22: Thank you for your suggestion. We have rewritten our conclusion for more consistent with our objective and explain what is the scientific development that this study can bring, Please see lines 413-419. 

Reviewer 2 Report

In manuscript "Genetic Effect and Growth Curve Parameter Estimation Under Heat Stress in Slow-Growing Thai Native Chickens" authors investigated three non-linear function to described growth curve of Thai Native Chicken and Thai synthesis chickens under heat stress condition and impacts of heat stress on the genetic absolute growth rate trait of these breeds. This is the first study on the genetic evaluation of heat stress in Thai native chicken. The manuscript was quite well written and presented, and it could be considered for publication after minor correction can be addressed (optional).

  1. The figure 1 is not really necessary because morphology or appearance of those breeds were not an objective of the study. Please remove it.
  2. L75-76: TSC is a crossbred between Chee males and broiler females. Broiler is not a breed,  so what is broiler breed?? And both TNC and TSC were considered as slow-growing breeds?
  3. L77: author said "This breeding strategy aims to promote growth performance and egg production" but the egg production was not investigated in this study.
  4. The data of the this study was collected over 5 years from 2015 to 2020, and the THI ranged from 72.6 to 81.36 while threshold of THI was 76.0. So how did authors collect the data in order to be sure that the data was collected when animals were under heat stress?
  5. Although threshold THI was 76, heat stress also can be classified in different extents such as mild, moderate, severe heat stress and the effect of each category on growth performance of chicken is different, does Gompertz function result in the same output if chicken were exposed in mild, moderate and severe?
  6. Except MSE and R2, are there any others statistical parameter from which we used to evaluate the best function to fit the growth curves of chickens?
  7. In the abstract and conclusion authors stated Gompertz function can be applied with the THI to evaluate genetic performance for heat tolerance and increase growth performance in slow-growing chicken, so by which way the function can increase growth performance of chickens?

Author Response

Response to Reviewer 2 Comments

In manuscript "Genetic Effect and Growth Curve Parameter Estimation Under Heat Stress in Slow-Growing Thai Native Chickens" authors investigated three non-linear function to described growth curve of Thai Native Chicken and Thai synthesis chickens under heat stress condition and impacts of heat stress on the genetic absolute growth rate trait of these breeds. This is the first study on the genetic evaluation of heat stress in Thai native chicken. The manuscript was quite well written and presented, and it could be considered for publication after minor correction can be addressed (optional).

Point 1: The figure 1 is not really necessary because morphology or appearance of those breeds were not an objective of the study. Please remove it.

Response 1: Indeed, the morphology or appearance of those breeds was not the study's objective. However, the presentation of chicken images in this study shows the uniqueness and differences in the appearance of the two chickens to understand and distinguish them. Therefore, we would like to ask permission to collect these two pictures for publication in this journal.

Point 2: L75-76: TSC is a crossbred between Chee males and broiler females. Broiler is not a breed, so what is broiler breed?? And both TNC and TSC were considered as slow-growing breeds?

Response 2: For TSC, the female broiler is a breed developed by the company, the Cobb 500 broiler, which is the co-investigator of this research. Therefore, while conducting research awaiting publication, it would be to ask for permission not to disclose detail of the broiler breed. Moreover, both TNC and TSC were considered slow-growing breeds because the ADG of both TNC and TSC was less than 50 grams/day, and this is the criterion for discriminating between slow-growing and fast-growing chicken.

Point 3: L77: author said "This breeding strategy aims to promote growth performance and egg production" but the egg production was not investigated in this study.

Response 3: We apologize for the misunderstanding; we have rewritten the sentence as follows: “The TSC breeding strategy aims to benefit from the complementary effect of both Thai native (good adaptability in harsh environments and good meat taste) and commercial broiler (high growth rate) chickens” Please see lines 82-84.

Point 4: The data of the this study was collected over 5 years from 2015 to 2020, and the THI ranged from 72.6 to 81.36 while threshold of THI was 76.0. So how did authors collect the data in order to be sure that the data was collected when animals were under heat stress?

Response 4: In fact, the weather data were recorded by a weather station near the experimental farm of Khon Kaen University so that air temperature and relative humidity were used to calculate THI values. In determining the origin of heat stress in this study (we still do not know), we analyzed the correlation between THI and growth data. The THI values were determined by levels starting from THI of 71 to THI 81, respectively. Finally, from statistical criteria were -2logL and AIC, we concluded that at THI of 76 was the starting point for heat stress in both chicken populations.

Point 5: Although threshold THI was 76, heat stress also can be classified in different extents such as mild, moderate, severe heat stress and the effect of each category on growth performance of chicken is different, does Gompertz function result in the same output if chicken were exposed in mild, moderate and severe?

Response 5: In this study, THI of 76 was considered mild heat stress (according to the THI table below), and hence the use of the Gompertz function was appropriate. However, for moderate and severe heat stress, we do not yet know the Gompertz function would be appropriate for these populations or not. The heat stress may become more moderate or severe; the growth curve function may also change.

Point 6: Except MSE and R2, are there any others statistical parameter from which we used to evaluate the best function to fit the growth curves of chickens?

Response 6: We agree with you because R2 is goodness-of-fit criteria and is appropriate for linear regression models, but in this study, the growth curve is a non-linear regression model; hence R2 may overestimate results. So, we changed from R2 to pseudo-R2 and added more statistical criteria as AIC because it can quantify the explanatory and predictive ability of the tested models and avoid data that may distort the results. For MSE values, we ask permission to use them. Please see Table 2.

Point 7: In the abstract and conclusion authors stated Gompertz function can be applied with the THI to evaluate genetic performance for heat tolerance and increase growth performance in slow-growing chicken, so by which way the function can increase growth performance of chickens?

Response 7: The Gompertz function was appropriate for fitting the growth curve in this study and used the genetic model (random regression model and THI function) in terms of AGR values (as of observation values). Therefore, the Gompertz function will improve body weight prediction accuracy at all stages of the chicken and lead to chickens with good growth genetics for the next generation. Moreover, AGR traits derived from a fitting model for use in genetic assessment combined with THI functions based on animal models repeatability test-day models. The EBV of the direct AGR and EBV of heat tolerance from the assessment were prioritized in the selection program.

Reviewer 3 Report

Dear authors,

Their work analyses the genetic effect of heat stress in growth curve of Thai Native chicken’s breed for the first time and it estimates different genetic parameters as variance components and heritability of absolute growth rate. The manuscript is well written and structured, the introduction provides sufficient background, the research design is appropriate, the methods are adequately described, the results are clearly presented, and the conclusions are supported by the results. From my point of view, the authors could expand the discussion of the results obtained, including studies where these parameters have been estimated in other breeds, or other characters in similar environmental conditions. In addition, authors should remove the underlining from "genetic lines" (lines 21 and 328), and they should unify the letter font in some paragraphs (line 330).

Author Response

Response to Reviewer 3 Comments

Dear authors,

Point 1: Their work analyses the genetic effect of heat stress in growth curve of Thai Native chicken’s breed for the first time and it estimates different genetic parameters as variance components and heritability of absolute growth rate. The manuscript is well written and structured, the introduction provides sufficient background, the research design is appropriate, the methods are adequately described, the results are clearly presented, and the conclusions are supported by the results. From my point of view, the authors could expand the discussion of the results obtained, including studies where these parameters have been estimated in other breeds, or other characters in similar environmental conditions. In addition, authors should remove the underlining from "genetic lines" (lines 21 and 328), and they should unify the letter font in some paragraphs (line 330).

Response 1: Thank you very much for your suggestion. We already removed the underlining from "genetic lines" in manuscript to "genetic lines" and format the font according to the instructions for Authors of Veterinary sciences journal. Moreover, we have rewritten our discussion section and added more information and study in other chickens. Please see in the discussion section.

Round 2

Reviewer 1 Report

Dear authors,

All the suggestions I made for the previous version of the article have been substantially improved. The text has been greatly improved and in addition, 13 new bibliographic references have been added to support the new information.

I want to congratulate you for the work done and I think that the manuscript has enough quality and interest to be published in "veterinary sciences".